



# Current and future wind energy resources in the North Sea according to CMIP6

Andrea N. Hahmann[1], Oscar García-Santiago[1], and Alfredo Peña[1]

[1]Department of Wind and Energy Systems, Technical University of Denmark, Roskilde, Denmark

**Correspondence:** Andrea N. Hahmann (ahah@dtu.dk)

**Abstract.** We explore the changes in wind energy resources in Northern Europe using output from historical to mid-twenty-first century CMIP6 simulations and the high-emission SSP5-8.5 scenario. This study improves upon many assumptions made in the past. First, we interpolate the winds to hub height using model-level raw data; second, we use a large ensemble of CMIP6 models; third, we consider the possible wake effects on the annual energy production of a large wind farm cluster proposed for the North Sea. The evaluation of the wind climatologies in the CMIP6 models over the North Sea for the historical period shows good correspondence with measurements from tall masts and three reanalysis data for 16 of the 18 models. Some of the models run at relatively high spatial resolution are as good as the reanalyses to represent the wind climate in this region. Our results show that annual mean wind speed and wind resources in Northern Europe are not particularly affected by climate change in 2031–2050 relative to 1995–2014, according to a sub-set of 16 models in the CMIP6 collection. However, the seasonal distribution of these resources is significantly altered. Most models agree on reductions in the future wind in summer in a band that extends from the British Isles to the Baltic Sea and on increases in winter in the Southern Baltic Sea. The energy production calculations show that summer energy production in a planned large wind farm cluster in the North Sea could be reduced by a median of $6.9\%$ during 2031–2050 when taking into account the wind farm wakes (that account by $-0.7\%$) and the changes in air density (that account by $-0.9\%$). Extrapolating 10-meter wind speeds to turbine height using the power law with a constant shear exponent is often a poor approximation. It can exaggerate the future changes in wind resources and ignore possible surface roughness and stability changes.

## 1 Introduction

The harvest of electricity from the wind plays a significant role among the climate change mitigation options. Integrated assessment models (i.e., models that link the main features of society and economy with the biosphere and atmosphere into one modelling framework) estimate that by 2050 renewable energy sources must supply $52$–$67\%$ of the primary energy to limit global warming to $1.5\,°C$ with no overshoot (IPCC, 2018). Wind and solar sources could provide nearly half of the required renewable energy (IPCC, 2018). However, wind and solar power plants supplied, respectively, only $6\%$ and $3\%$ of



the global electricity consumption in 2020 (IRENA, 2019). Fortunately, cost reductions in wind power technologies and wind

farm construction have increased the economic attractiveness of wind farms in many regions (IRENA, 2019; Beiter et al.,
2021; Wiser et al., 2021). Therefore, because of its mitigation potential and lower costs, the construction of new wind farms
will likely accelerate in the coming decades.

  However, climate change could impact the supply of renewable energy, including wind energy (Cronin et al., 2018; Yalew
et al., 2020; Gernaat et al., 2021). Many countries currently plan significant wind power expansion with global projections for

installed wind capacity in 2040 of $2400$–$3320\,\mathrm{GW}$ onshore (from about 700 GW in 2020) and 342–562 GW offshore (from
about 34 GW in 2020) (IEA, 2019; GWEC, 2020, 2021). As one particular example, in 2021, the Danish Parliament approved
the construction of two energy islands: one offshore near the natural island of Bornholm in the Baltic Sea and a second as an
artificial island in the North Sea. The turbines off the coast of Bornholm will have a capacity of $2\,\mathrm{GW}$. The proposed North
Sea Energy Island wind farm cluster will become one of the largest energy hubs in Europe, with a target of $10\,\mathrm{GW}$ installed

capacity in 2050 (COWI, 2020). The energy production and the financial profitability of all new planned wind farms, including
the two Energy Island clusters, are calculated using data from past historical wind resources (COWI, 2020). Still, there is a
probable threat that wind resources will change in the future due to climate change during the lifetime of a wind farm.

  The study of future changes in wind resources is not a new subject. A systematic literature search conducted in early 2022
with the keywords "Wind Resources" and "Climate Change" returned nearly 90 peer-reviewed articles from 2005. These

include the assessment of future wind resources at the global (Karnauskas et al., 2018; Zheng et al., 2019), regional (Devis
et al., 2018; Chen, 2020) and local scale (Chang et al., 2015; Alonso Díaz et al., 2019). The most recent papers that study the
projected changes in wind resources use the wind speed simulated by global climate models from either the Coupled Model
Intercomparison Project Version 5 (CMIP5; Taylor et al., 2012), the newer Version 6 (CMIP6; Eyring et al., 2016a), or regional
climate models such as CORDEX (Giorgi and Gutowski, 2015). The CMIP5/6 and CORDEX models have a spatial resolution

of hundreds or many tens of kilometres, respectively. All the studies, except for two (Sailor et al., 2008; Wang et al., 2020),
used the raw wind speed output from the climate model to estimate the power production of a single wind turbine and explore
how this will change in the future. The raw climate model output over land nearly always underestimates the extractable
wind resources because it does not correctly consider the full spectrum of atmospheric motions that modulate the wind near
the Earth's surface. In addition, wind turbines and wind farms "shade" each other at a range of spatial scales when nearby

(Barthelmie and Jensen, 2010; Nygaard, 2014). Wake effects are also ignored in all previous studies.

  Earlier studies have made many assumptions that do not hold when studying real-life wind farms. First, $60\,\%$ of the arti-
cles about future wind resources use the 10-m wind output from the climate model to diagnose the hub height ($\approx$50–200m)
wind speed. About half of these use a power law with a constant shear exponent of 1/7 for land and offshore areas. Such an
approximation generally overestimates the hub-height wind speed over the ocean. It exaggerates future wind speed changes

because such a shear exponent is typical of winds over grass areas at $\approx$30–50 m. Even when an offshore value of the shear
exponent is used for wind extrapolation over the sea, the value is constant, ignoring that air temperature will likely increase
and atmospheric stability could vary in time. Second, the aerodynamic characteristics of the land surface and their changes in
space and time are also of utmost importance for wind resource assessment. For example, changes in wind direction, coupled



or uncoupled with changes in land use, could affect wind resources by changing the relationship between the orientation of the dominant flow and the orientation of the orography and the coastline. This effect can be ignored offshore and away from coastlines but could be very important over the land (Badger et al., 2022). Third, a large portion of the previously mentioned articles used a small sample of climate models. For near-surface wind-related quantities, models often disagree on the sign of future wind changes (Karnauskas et al., 2018; Pryor et al., 2020) thus, useful predictions should rely on as many models as possible. Fourth, all published studies disregard the effect of potential changes in wind direction and how these changes interact with the layout of ever-larger wind farms and their wake losses. Most previous studies use the power curve from a single wind turbine disregarding any wake effects. Lastly, most of the studies identify the impact of climate change on wind resources at periods far in the future, 2050 or 2100. This is relevant for theoretical studies, for example, Integrated assessment models (Gernaat et al., 2021), but governments and wind farm developers are asking these questions for the near future, 20–30 years from now.

This study improves upon four of the five assumptions described above. First, we interpolate the winds to hub-heights using model-level raw data. Second, we use all available CMIP6 models, which have all the necessary fields for the vertical interpolation of the wind speed at the highest output frequency (6 hours). Third, we consider a huge wind farm proposed for the North Sea and investigate the role of the future changes in wind direction and wake losses. Finally, we assess climate change for the 20 years 2031–2050, not at the end of the century. We concentrate our study on a relatively small area of Northern Europe, but this is where large wind farms already operate, additional development is planned, and data from tall offshore masts are available for verification of the winds in the reanalysis and climate models in the historical period. We ultimately demonstrate how a study of future climate changes in wind resources could be applied to a relatively small area and be helpful to energy planners and wind farm developers. The methods demonstrated here can be easily transferred to any other region offshore.

The paper is organised as follows. Section 2 describes the data from climate models, reanalyses, and tall masts used in the study. It also compares masts and reanalyses data to decide on an observed reference for the wind climate of the past. Section 3 describes the vertical interpolation and the verification metrics used in the study. The characteristics of the wind farm cluster used in the last part of the study are described in Sec. 3.3. We evaluate the various reanalysis against tall mast data and reanalyses against CMIP6 models for the historical period in Sec. 4. The results of the evaluation of wind resources in the future are shown in Sec. 5 for the North Sea. Section 6 presents the analysis for the future Energy Island wind farm cluster. We finalise with a discussion of the results in Sec. 7 and conclusions in Sec. 8.

## 2 Data

### 2.1 CMIP6 models, simulations and data

Twenty years ago, the CMIP project started as a platform to compare early coupled climate models (Meehl et al., 1997). The models being coupled were the global atmosphere, dynamic ocean and thermodynamic sea ice, and simple land surface models. These early models often required "flux adjustments" to keep their simulated climate from drifting when performing decade-long time integrations. The models used in CMIP6 have been improved, no longer using flux adjustments, and incorporate many





other sub-models, including, for example, river runoff, sea ice dynamics, ocean waves, atmospheric chemistry, and models that explicitly predict natural $CO_2$ concentrations via bio-geophysical cycles in the land and ocean biosphere. These sub-models are needed to carry on the CMIP6 project design to understand the response of the Earth's system to forcing and how we can
assess future climate changes given the internal climate variability, predictability, and uncertainties in scenarios (Eyring et al., 2016b).

Table 1 describes the main characteristics of the CMIP6 models used in this study. Two general types of models are used in CMIP6 simulations that differ in treating natural $CO_2$ concentrations. Earth System Models (ESMs) account for the fluxes of $CO_2$ between the atmosphere, ocean, and biosphere. In Climate Models (CM, also known as Atmosphere-Ocean General
Circulation Models, AOGCMs), changes in $CO_2$ concentrations are prescribed and cannot vary in response to climate change. In both types, anthropogenic sources of $CO_2$ and other Greenhouse Gases (GHGs) are prescribed. We use data from both CMs and EMSs with spatial resolutions that vary from coarse ($2.81°$) to relatively fine ($0.83°$ in longitude).

The historical forcings are based, as far as possible, on observations and cover the period 1850–2014. These include emissions of short-lived species and long-lived GHGs, GHG concentrations, global gridded land use, solar forcing, and stratospheric
aerosols (e.g., volcano emissions). For models without ozone chemistry, time-varying gridded ozone concentrations and nitrogen deposition are also provided. For future climate simulations (2015–2100), CMIP6 also provided coordinated multi-model climate projections based on alternative scenarios of future emissions and land-use changes produced with integrated assessment models (O'Neill et al., 2016), which, in turn, are determined from future pathways of societal development, i.e., the Shared Socioeconomic Pathways (SSPs) (Riahi et al., 2017). The historical and future global climate model data were ob-
tained from the CMIP6 DECK experiments; we use data from the *historical* simulations for the period 1980–2014 (1850–2014 is available for many models), and the *SSP5-8.5* scenario for the period 2015–2050 (2015–2100 is available). In total, we downloaded a 70-year-long time series for all models.

We use data from climate simulations forced by the high end of the range of future pathways emissions, i.e., SSP5-8.5, with high enough anthropogenic emissions to produce a radiative forcing of $8.5\,\mathrm{W\,m^{-2}}$ in 2100. We chose to use this extreme
scenario for two reasons. First, it might demonstrate the maximum expected impact on wind resources, and second, it is the scenario with the most available data needed for this study. In the CMIP6 SSP5-8.5 scenario simulations, the forcing is specified as land use (fractions of crops, pasture, urban, and forest areas), emissions and concentrations of long-lived GHGs, air pollutants emissions, and short-lived forcing by gases (e.g., ozone). We use data from the near future, 2015–2050 because it could influence the planning and building of wind farms in the coming years. Some past studies have supplied changes for
times further in the future, e.g., 2071–2100 (Jerez et al., 2019; Chen, 2020). However, wind farm developers are currently planning these farms and need data useful to assess the short-term future, i.e., during the expected lifetime of a wind farm.

The data used in this study are a particular set from the "6hrLev" table, which contains time series of zonal and meridional winds, $u$ and $v$, respectively, air temperature ($T$), and specific humidity ($q$) on the original model levels and surface pressure ($p_s$) every 6 hours. The data are available from the Earth System Grid Federation (ESGF; Cinquini et al., 2014) database
downloaded via OPeNDAP. To locate the desired dataset, we use the Python esgf-pyclient package (esg, 2016). The vertical structure of most models warranties that wind data exist for levels below 100 m above ground level (AGL) for all models and



below 50 m AGL for 14 of the 18 models. For this reason, we use winds at 100 m AGL, which is close to modern offshore wind farm hub heights.

**Table 1.** Models in the CMIP6 archive with $u, v, T$, and $q$ available at model levels and 6-hourly output in the **historical** and **SSP5-8.5** simulations. CM: Climate Model, ESM: Earth System Model.

| Model name | Modeling center (Country) | Model type | Horizontal grid spacing[a] (lon × lat) | Total number vertical levels (heights below 250 m AGL) | Citation |
|---|---|---|---|---|---|
| ACCESS-CM2 | CSIRO (Australia) | CM | $1.25° \times 1.875°$ | 85[b] (10.0, 36.7, 76.7, 130.0, 197.0) | Tilo et al. (2020) |
| CanESM5 | CCCma (Canada) | ESM | $2.8125° \times 2.79°$ | 49 (38.1, 107.1, 215.5) | Swart et al. (2019) |
| CESM2 | NCAR (USA) | ESM | $1.25° \times 0.94°$ | 32 (60.0, 193.3) | Danabasoglu et al. (2020) |
| CMCC-CM2-SR5 | CMCC (Italy) | CM | $1.25° \times 0.94°$ | 30 (57.7, 185.1) | Cherchi et al. (2019) |
| CNRM-CM6-1 | CNRM (France) | CM | $1.4° \times 1.4°$ | 91 (9.3, 31.7, 62.9, 104.1,155.3, 215.8) | Voldoire et al. (2019) |
| CNRM-ESM2-1 | CNRM (France) | ESM | $1.4° \times 1.4°$ | 91 (9.5, 32.6, 64.6, 106.9, 159.6, 221.9) | Séférian et al. (2019) |
| HadGEM3-GC31-LL | UKMO (UK) | CM | $1.875° \times 1.25°$ | 85[b] (10., 36.7, 76.7, 130.0, 196.7) | Sellar et al. (2020) |
| HadGEM3-GC31-MM | UKMO (UK) | CM | $0.833° \times 0.556°$ | 85[b] (10., 36.7, 76.7, 130.0, 196.7) | Sellar et al. (2020) |
| IPSL-CM6A-LR | IPSL (France) | CM | $2.5° \times 1.27°$ | 91 (9.3, 29.1, 51.3, 76.3, 104.3, 135.6, 170.4, 209.4) | Boucher et al. (2020) |
| MIROC6 | MRI (Japan) | CM | $1.4° \times 1.4°$ | 81 (19.8, 67.4, 135.2, 223.7) | Tatebe et al. (2019) |
| MIROC-ES2L | MRI (Japan) | ESM | $2.8125° \times 2.79°$ | 40 (19.8, 67.5, 135.4, 224.0) | Hajima et al. (2020) |
| MPI-ESM1-2-LR | MPI (Germany) | ESM | $1.875° \times 1.865°$ | 47 (30.5, 138.2) | Mauritsen et al. (2019) |
| MPI-ESM1-2-HR | MPI (Germany) | ESM | $0.9375° \times 0.935°$ | 85 (29.7, 134.5) | Müller et al. (2018) |
| MRI-ESM2-0 | MRI (Japan) | EMS | $1.125° \times 1.121°$ | 80 (11.5, 34.5, 65.4, 112.1, 178.9) | Kawai et al. (2019) |
| NESM3 | NUIST (China) | EMS | $1.875° \times 1.865°$ | 47 (30.7, 139.0) | Yang et al. (2020) |
| NorESM2-LM | NorESM (Norway) | EMS | $2.5° \times 1.895°$ | 32 (59.4, 190.5) | Seland et al. (2020) |
| NorESM2-MM | NorESM (Norway) | EMS | $1.25° \times 0.942°$ | 32 (59.6, 191.3) | Seland et al. (2020) |
| UKESM1-0-LL | UKMO (UK) | EMS | $1.875° \times 1.25°$ | 85[b] (20.0, 53.3, 100.0, 160.0, 233.3) | Sellar et al. (2020) |

[a] Approximate value in the region because some models use Gaussian grid with variable latitude grid spacing. [b] Model data already in height above ground level.





## 2.2 Mast observations and reanalysis

We use data from six tall masts (Fig. 1a), with relatively long duration, in the North and Baltic Sea to evaluate the boundary layer winds in the reanalysis datasets. The data are from offshore platforms (FINO1, FINO2, FINO3, IJmuiden and Ekofisk), a coastal met mast (Høvsøre) and are down-sampled to hourly from their original 10-minute averages. When the site has various measurement levels, we select the highest and less disturbed as described in Hahmann et al. (2020). The processing of the data from the mast on the oil platform Ekofisk is described in Solbrekke et al. (2020). We use the data from the $102\,\mathrm{m}$. Figure 1a

also shows the location of an averaging box (54–57.5°N and 1–7.5°W) used later in the future climate calculations and the location of the proposed Energy Island wind farm cluster.

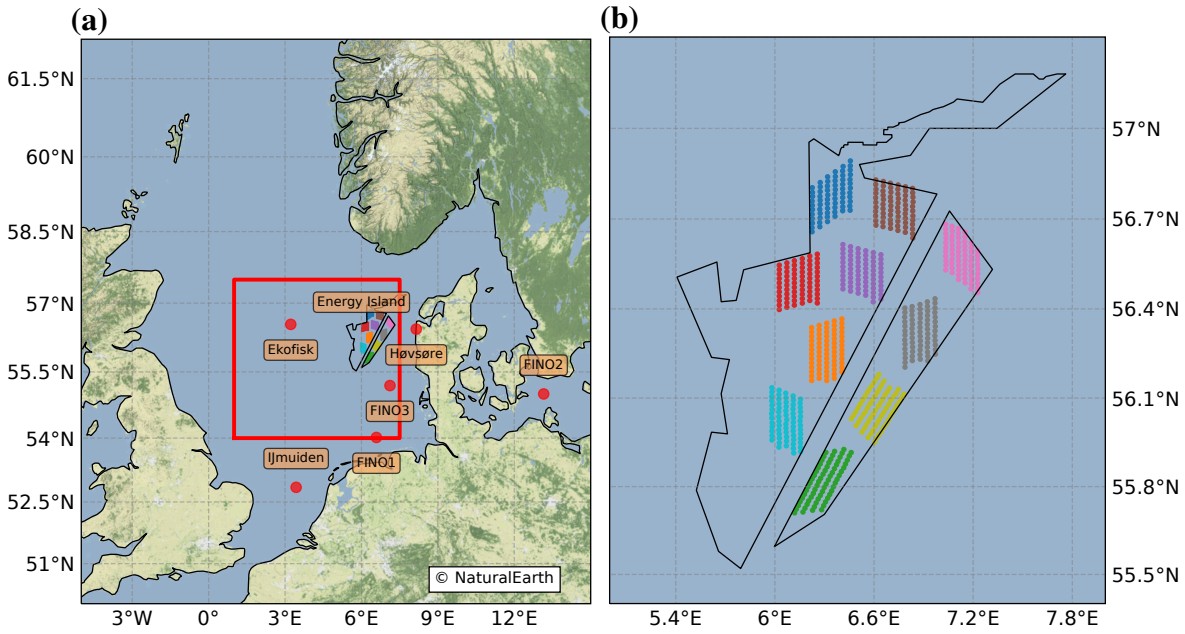

**Figure 1.** (a) Location of the sites used for validation (red markers) and the area used for spatial averaging (red box). The area in black represents the layout of the Energy Island wind farm cluster. (b) Layout of the future Energy Island wind farm cluster area (black lines) and the ten wind farms in markers with different colours.

We use the wind speed and direction derived from four modern reanalysis: ECMWF ERA5 (Hersbach et al., 2020); NASA Modern-Era Retrospective Analysis for Research and Applications, Version 2 (MERRA2; Gelaro et al., 2017); NOAA Twentieth Century Reanalysis version 3 (20CR; Slivinski et al., 2019); and the New European Wind Atlas (NEWA; Dörenkämper

et al., 2020). Some general details for each reanalysis are listed in Table 2, including their advantages and disadvantages from a wind energy perspective. The wind speeds and directions from MERRA2 are interpolated to $100\,\mathrm{m}$ AGL from the native model levels, similarly as done with the CMIP6 data (this is described in Sec. 3.1).



**Table 2.** Description of the various modern reanalysis datasets used in the model evaluation of the historical wind climate.

| reanalysis product (release year) | spatial resolution (number of levels) | output frequency | time period | advantages | disadvantages |
|---|---|---|---|---|---|
| NEWA (2019) | 3 km × 3 km (62) | 30 min | 1989–2018 | very high resolution; tailored for wind energy | no longer updated; no assimilated data |
| ECMWF ERA5 (2016) | $0.25° \times 0.25°$(137) | 1 hour | 1979– | high resolution; 100-m wind directly available | uses sub-grid orographic drag |
| NASA MERRA2 (2015) | $0.5° \times 0.625°$(72) | 1 or 3 hours[a] | 1980– | updated often | medium resolution; only 50-m wind directly available |
| NOAA 20CRv3 (2019) | $1° \times 1°$(28) | 3 hours | 1850– | long duration; consistent assimilated data | low spatial and temporal resolution |

[a] MERRA2 50 m AGL winds are available every 1 hour and atmospheric fields every 3 hours.

## 3 Methods

### 3.1 Vertical interpolation of winds

The CMIP6 data from the various models are available via the ESGF database. The `6hrLev` table contains the high-temporal resolution atmospheric model data on pressure-sigma levels or heights above the model terrain. Thus, the wind speed and direction can be interpolated to any height above the first model level, which is listed in Table 1. For the models with sigma coordinates, the pressure at model level $k$ is computed from

$$p_k = a_k\, p_0 + b_k\, p_s, \tag{1}$$

where, $p_0$, $a_k$ and $b_k$ are the reference pressure and the sigma level coefficients at level $k$ and $p_s$ is the surface pressure. The thickness between two model layers, $z_{k+1}$ and $z_k$, can be determined from the hypsometric equation:

$$\Delta z = z_{k+1} - z_k = \frac{R_d \overline{T_v}}{g} \ln\left(\frac{p_k}{p_{k+1}}\right), \tag{2}$$

where $T_v = T\,(q+\epsilon)/[\epsilon\,(1+q)]$ is the virtual temperature, which is averaged (therefore the overbar) between levels $k$ and $k+1$, $\epsilon = 0.622$, $R_d$ is the gas constant for dry air, and $g$ is the Earth's gravitational constant; $T$ and $q$ are the air temperature (`ta`) and

specific humidity (`hus`). The height of each model level can be obtained by integrating Eq. (2) from $p = p_s$ to the model level pressure $p = p_k$. Knowing the height of each model level, the horizontal wind speed, $U = \sqrt{u^2 + v^2}$, can then be interpolated to any height above the model terrain. $u$ and $v$ are the two wind components, named `ua` and `va` in the ESGF database. We use a linear interpolation in $\ln z$ for three heights, $h = 50, 100$, and $200$ m AGL. To obtain the wind direction $\phi$ at these heights, we linearly interpolate the two horizontal wind components in height separately and recalculate the resulting wind direction. The

evaluation of the past and future wind climate are for a hub height of $100\,\mathrm{m}$ AGL.





The wind power density $P$ is defined as

$$P = \frac{1}{2}\rho U^3, \tag{3}$$

where $\rho = p_s/(R_d T_{vs})$ is the surface air density and $T_{vs}$ is the surface virtual temperature. Sometimes we use the air density for the standard atmosphere, $\rho_{st} = 1.225\,\mathrm{kg\,m^{-3}}$. We name the wind power density computed using the standard air density the standard wind power density $P_{st}$.

We also use the 10-m wind speed directly from the ESGF archive for comparisons. This field is named `sfcWind` in the daily and monthly data archive. As in previous studies, we use the power-law to extrapolate from 10 to $100\,\mathrm{m}$ AGL,

$$U_{\mathrm{PL,100\,m}} = U_{10\,\mathrm{m}}\Big(\frac{100\,\mathrm{m}}{10\,\mathrm{m}}\Big)^{\alpha}, \tag{4}$$

where $\alpha = 1/7$. The implications of using the power law to extrapolate winds from 10 m instead of the vertical interpolation of wind speed from model levels is discussed later in the paper.

### 3.2 Model validation metrics

We use several metrics to evaluate the model simulations' accuracy compared to atmospheric reanalysis. The bias is a popular error statistic for comparing the wind speed distributions between observations and model-simulated fields. However, since the power density is a function of the cube of the wind speed (see Eq. 3), the shape of the wind speed distribution is more critical. Small changes in the wind speed distribution are amplified when converted to power, especially in the upper end of the distribution. Accordingly, we use the Earth Mover's Distance (EMD) to evaluate the differences in the shape of two frequency distributions (Hahmann et al., 2020). The EMD is equivalent to the area between the two cumulative distribution functions for one-dimensional distributions. The circular EMD (CEMD; Rabin et al., 2008) extends the EMD concept to one-dimensional circular histograms, such as the frequency distribution of wind directions. The smaller the value of EMD and CEMD, the better the distribution of the simulated wind speed and direction match those observed.

To diagnose the ability of the models to represent the annual cycle of wind speed, we compute the monthly mean of wind speed and then their average over the historical period ($\overline{U}_o^i$ and $\overline{U}_m^i$, where the underscores $o$ and $m$ stand for the observed and modelled values and $i = 1...N$ is the month). To evaluate the annual cycle in wind speed in the CMIP6 model against that of the reanalysis, we compute the root mean square difference (RMSD) as

$$\mathrm{RMSD} = \Big[\frac{1}{N}\sum_{i=1}^{N}(\overline{U}_m^i - \overline{U}_o^i)^2\Big]^{1/2}, \tag{5}$$

where $N = 12$ months. We also use the EMD, CEMD and RMSD to diagnose changes in the future wind climatology compared to historical conditions.

We use boxplots to compare wind speed distributions and other parameters among sites, reanalysis, and CMIP6 models. The boxplots in this paper show the "minimum", first quartile (Q1), median, third quartile (Q3), and "maximum" of each distribution. The minimum and maximum are defined as Q1 $-1.5\,\mathrm{IQR}$ and Q3 $+1.5\,\mathrm{IQR}$, respectively, and IQR = Q3 $-$ Q1 is the interquartile range.





## 3.3 Wind farm energy production calculations

We perform annual or seasonal energy production calculations for the planned Energy Island wind farm cluster in the North Sea (Fig. 1b) using the time series of the past or the future wind speed and direction derived from the CMIP6 models and the reanalyses. The planned wind farm cluster consists of ten wind farms, each having an installed capacity of $1\,\text{GW}$. The wind farm layouts (using a spacing of $7 \times 12$ rotor diameters D with D = $240\,\text{m}$) in Fig. 1b come from the Energy Island report (COWI, 2020), consisting of 67 $15\,\text{MW}$ wind turbines for each wind farm layout. The turbine locations in each wind farm were hand-digitalized.

We use the engineering wake model of Jensen as described in Göçmen et al. (2016) for wake calculations. This wake model accounts for the momentum (or velocity) deficit, which causes a reduction in the power output of the downstream turbines. The wake model superposes the individual turbine wakes by considering the wind direction and wake length. This wake model does not account for the mesoscale shadowing of one wind farm to another farm downstream. The wake model needs information on the thrust curve, rotor diameter, and the wake expansion parameter, $k$. The wake expansion is an empirical parameter that varies according to the wind farm, surface roughness length, and averaged atmospheric turbulence (Peña and Rathmann, 2014; Peña et al., 2016). We set the Jensen wake expansion parameter $k$ to $0.04$, although this depends on turbulence, among others (Peña and Rathmann, 2014; Peña et al., 2016). We use the IEA $15\,\text{MW}$ reference wind turbine (Gaertner et al., 2020) for the wind turbine characteristics. The wind turbines have a hub height of $150\,\text{m}$.

The power output for the wind turbines in the farm cluster is calculated every 6 hours using the wind speed and wind direction from the ensemble of CMPI6 models and three reanalyses for the ten wind farms. We also calculate the power time series at the highest frequency available for ERA5 ($1\,\text{h}$), MERRA2, and 20CR ($3\,\text{h}$) reanalyses to estimate the effect of the aggregation to 6 hours in the CMIP6 models. The wake-free wind speed and direction time series, which are input into the wake model, are derived by vertical interpolation to hub height ($z = 150\,\text{m}$) and horizontal interpolation to the coordinates of the centre of the Energy Island cluster (56.4°N, 6.0°E, Fig. 1b). For simplicity and because of the smooth nature of the CMIP6 model output, we assume the wind speed and direction as horizontally homogeneous through the wind farm cluster, and so, it is only perturbed by the turbines' wakes.

The energy production has units of $\text{GW}\,\text{h}$. It is calculated by summing up the 6-hours energy output over all turbines within the analysed period or over a particular season within the analysed period. This also assumes that the energy output is constant over the 6 hours. We perform three possible energy production calculations for the historical (1995–2014) and future (2031–2050) periods. The reference energy production, $\text{EP}_{ref}$, which uses the nominal turbine power curve by multiplying it by the total number of wind turbines (no wake losses are included). The "standard EP", $\text{EP}_{st}$, which includes wake loses but uses the standard density value. And finally, the net EP considers both the wake losses and variable air density. Wind farm yield calculations often include other technical losses such as availability of wind farms (some turbines may be offline due to faults), electrical efficiency, turbine performance, and curtailment. These losses are ignored in this study.



## 4 Evaluation of the historical CMIP6 wind climatologies

Since the mast measurements are at different heights, cover different periods and have various output frequencies than the CMIP6-derived winds, we evaluate the wind climatology from the CMIP6 historical simulations using a multi-step approach. We first compare the time series of mast measurements to reanalysis at the six sites at the highest possible time resolution. Once we demonstrate that these reanalyses are adequate, the CMIP6 evaluation is carried out using only the 6-hourly reanalysis data at the centroid of the Energy Island at a single level, 100 m AGL, but for the entire historical period (1980–2014), considering

the wind speed and wind direction distribution, and the amplitude of the annual cycle in wind speed. To complete the wind climate evaluation, we survey the summary statistics (Sect. 3.2) for the sites in the North Sea and the Energy Island location. Lastly, we present the comparison of the EP calculations for the Energy Island.

Figure 2 shows the evaluation of the four different reanalysis (Table 2) against the observations for the six sites (Sect. 2.2). The reanalysis wind fields have been bi-linearly interpolated to the exact coordinates and height of the tall mast measurements.

The data availability varies among the sites: from 15 years at Ekofisk to 4.5 years at IJmuiden. As illustrated, the four reanalyses represent the wind climatology very well both in terms of the mean wind speed (absolute biases $< 0.5\,\mathrm{m\,s^{-1}}$ at 75% of the sites and reanalysis) and its distribution, except for 20CR. At the three sites that surround the Energy Island, Høvsøre, FINO3, and Ekofisk, the bias between the wind speed estimated by the ERA5 reanalysis and that observed are $0.04\,\%$, $-2.81\,\%$, and $-5.69\,\%$, respectively. The 20CR wind speeds are lower than the observed wind speeds at all sites, especially at FINO2, which

is influenced by the proximity to land in the Baltic Sea. To evaluate the CMIP6 simulations during the historical period, we continue with ERA5, MERRA2 and 20CR with a grid spacing comparable to the climate models.

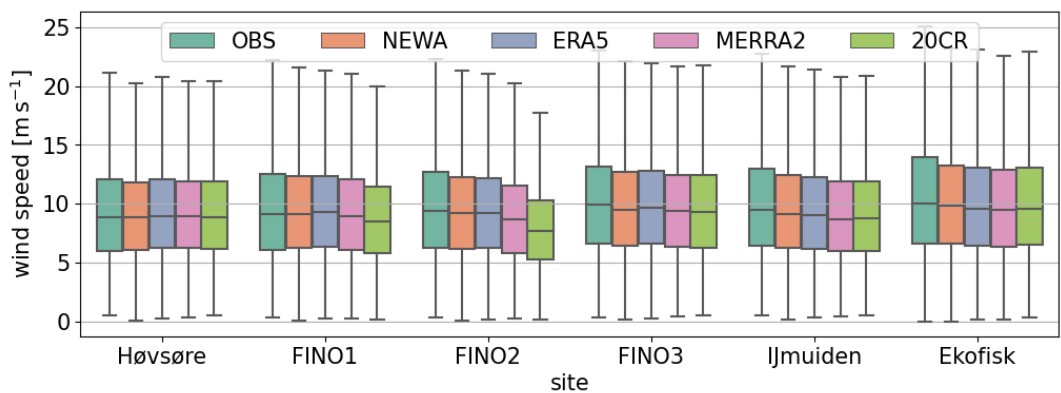

**Figure 2.** Comparison of the observed and reanalysis-derived wind speed statistics at the six observed sites in Fig. 1a. The reanalysis and observations are time synchronised and thus are at a 3 hour frequency. The boxplots show the "minimum", first quartile (Q1), median, third quartile (Q3), and "maximum" of each distribution.



## 4.1 The wind speed climatology at the Energy Island location

Figure 3 compares the 100-m wind speed climatologies among the CMIP6 models and the reanalysis for the time series at the centroid of the proposed Energy Island (Fig. 1) for the full historical period (1980–2014). As previously demonstrated when comparing to the observed data (Fig. 2), the wind climate of the various reanalysis is nearly identical, with a mean of 10.0–10.2 m s$^{-1}$. The wind climatology of the CMIP6 models is comparable to that of the reanalysis models, with mean values in the range of 9.1–10.7 m s$^{-1}$. The exception is the two simulations using the MIROC model, with a mean of 8.1 m s$^{-1}$ and 6.3 m s$^{-1}$, which is well outside the other models' range; it is not clear whether the MIROC model discrepancies are due to problems in the model simulation itself or the data provided in the ESGF archive. Comparing the 10-m wind speed for the two MIROC models against that of the other CMIP6 models (not shown) does not display such a large discrepancy. The median of most models is within ±1.5 m s$^{-1}$ of that estimated by the three large-scale reanalyses. In general, the IQR in the CMIP6 models is slightly larger than that estimated by the reanalysis. The maximum wind speed, excluding any outliers, in the CMIP6 models is 20–25 m s$^{-1}$, consistent within about 1–3 m s$^{-1}$ with the three reanalyses, except for the MIROC and CanESM5 models.

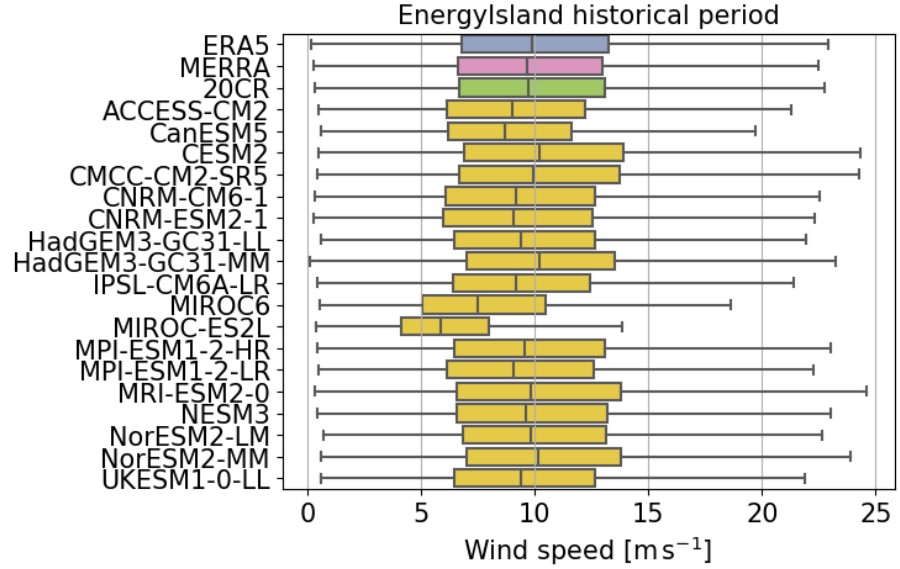

**Figure 3.** Comparison of the wind speed at 100 m AGL simulated by the CMIP6 models (see Table 1) and that provided by global reanalyses during the full historical period (1980–2014) for the Energy Island. The 6-hourly data from each CMIP6 model is synchronised with the ERA5 reanalysis. The boxplots show the "minimum", first quartile (Q1), median, third quartile (Q3), and "maximum" of each distribution.

We also compare the combined wind speed and direction distribution (wind roses) for two CMIP6 models (i.e., the best and the worst) to the ERA5 reanalysis for the Energy Island. Figure 4 shows a remarkable similarity between the wind rose of ERA5 and the MPI-ESM1-2-HR model (CEMD = 3.8°), but poor similarity between ERA5 and the coarse resolution NorESM2-LM





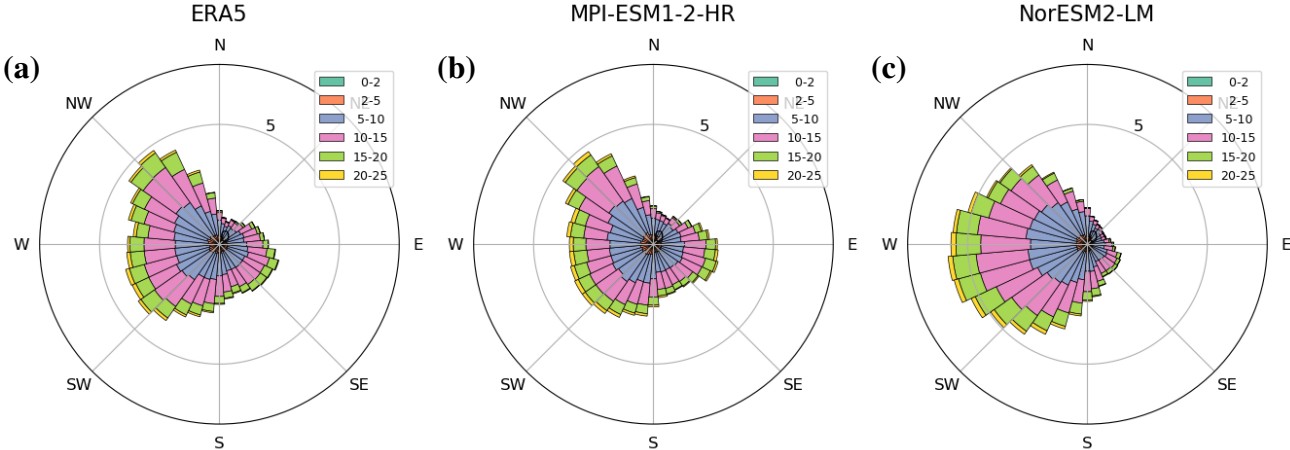

**Figure 4.** Wind roses of the wind climatology at 100 m AGL in the (a) ERA5 reanalysis and simulated by the (b) MPI-ESM1-2-HR and (c) NorESM2-LM CMIP6 models during the historical period (1980–2014) at the Energy Island. Wind speed bins are in $\mathrm{m\,s^{-1}}$.

model (CEMD = $14.8°$). All models represent the dominant SW to NW flow in various degrees (not shown). Further analysis for all models and sites is done later in this section (see Fig. 6c for all CEMD for all models).

Figure 5 shows the annual cycle of the monthly mean wind speed at $100\,\mathrm{m}$ AGL from the CMIP6 models compared to the three reanalyses for the historical period at the Energy Island. The monthly means of nearly all models are primarily within one standard deviation of the monthly mean of the ERA5 reanalysis (blue shaded area). The three CMIP6 models that significantly fall outside this region are the same models previously flagged in Fig. 3: the two MIROC and CanESM5 model simulations. While the monthly wind speed evolution simulated by these three models follows the phase and amplitude of the reanalysis,

their monthly mean wind speeds are too low. The monthly mean wind speeds simulated by the CanESM5 are too low only during the summer and autumn months but within the variability of the ERA5 monthly means during the rest of the year.

### 4.2    Evaluation at multiple sites

We carry on with the evaluation of the wind climatology in the CMIP6 models at multiple sites in the North Sea. We use summary heat tables (Fig. 6) to compare the wind climate in the CMIP6 models to that of the ERA5 at 100 m AGL, covering

the entire historic period, 1980–2014. On average, $2/3$ of the models underestimate the mean wind speed when compared to ERA5, especially on the east side of the basin (Fig. 6a). At the Ekofisk and IJmuiden sites in the central and southern North Sea, the bias in the mean wind speed compared to ERA5 is more positive; only four CMIP6 model simulations, including those by the MIROC model, overestimate the mean wind speed at Ekofisk. The results are similar among models; when they have small BIAS, they also have small EMD, CEMD and the annual cycle RMSD and vice versa. The only anomaly is the NorESM2-

LM model, which has reasonable BIAS, EMD, and RMSD, but large deviations in the wind rose as diagnosed by the CEMD (Fig. 6c). These statistics do not detect a better-simulated wind climatology in the CMIP6 CMs than in the ESMs. However,



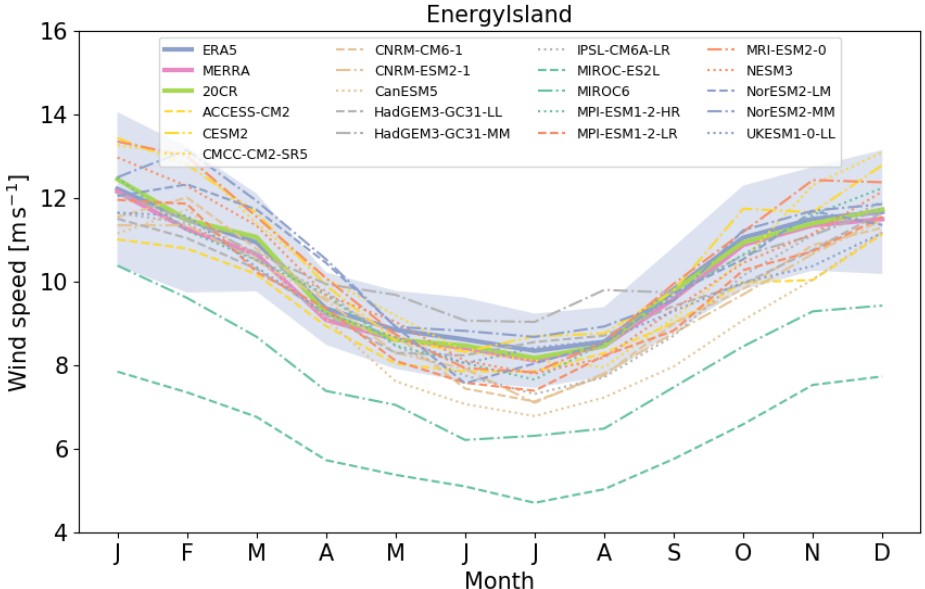

**Figure 5.** The annual cycle of mean monthly wind speed ($\overline{U}_o^i$ and $\overline{U}_m^i$) at 100 m AGL simulated by the CMIP6 models and that provided by the global reanalyses during the full historical period (1980–2014) for the Energy Island location. The shaded area is the $\pm$ of one standard deviation of the monthly means for the ERA5 reanalysis.

high spatial resolution seems advantageous for wind direction. For example, the CEMD statistics are $2.6–5.2°$ for the $1.875°$ HadGEM3-GC31-LL model but $2.0–3.9°$ for the $0.833°$ HadGEM3-GC31-LL model simulation. This is also true for the MPI-ESM1-2-HR (CEMD = $1.6–3.7°$) versus MPI-ESM1-2-LR (CEMD = $3.9–5.6°$). Based on all four metrics, the simulation using the HadGEM3-GC31-MM model is as good as that using the other two reanalyses. **?** reached similar conclusions regarding the improvement of the atmospheric circulation in the CMIP6 models.

### 4.3 Annual energy production at the Energy Island cluster

As described in Sec. 3.3, we perform Annual Energy Production (AEP) calculations for the hypothetical wind farm cluster at the Energy Island. Figure 7 compares the net AEP from the wind farm cluster computed from the wind climate of the reanalysis and the CMIP6 models during the entire historical period 1980–2014. The net AEP values from the MIROC models are nearly half of the other models and are not included in the figure. The mean net AEP for the Energy Island wind farm cluster calculated using the reanalyses is between $56–57\,\mathrm{TWh}$. The use of 6-hourly data does not dramatically influence the calculation; if one uses hourly or 3-hourly data in the ERA5 time series, the net AEP is nearly identical and only $0.03\,\%$ larger than using the 6-hourly time series, respectively. The wind farm cluster is composed of $670\ 15\,\mathrm{MW}$ wind turbines; thus, the capacity factor is about 0.64 when using the ERA5 data. The capacity factor estimate is slightly more optimistic than that from COWI for three of the wind farms in the Energy Island cluster, which is about 59% ($5200\,\mathrm{GWh}$ per wind farm according to COWI, 2020).



**Figure 6.** Evaluation metrics of the 100-m wind climatologies for the reanalysis and CMIP6 models, i.e., (a) BIAS $[\mathrm{m\,s^{-1}}]$, (b) EMD $[\mathrm{m\,s^{-1}}]$ for the simulated wind speed, (c) CEMD $[°]$ for simulated wind direction and (d) RMSD of the mean annual cycle $[\mathrm{m\,s^{-1}}]$, at the six locations in the North Sea for the full historical period (1980–2014). The wind climatologies are evaluated against that of the ERA5.





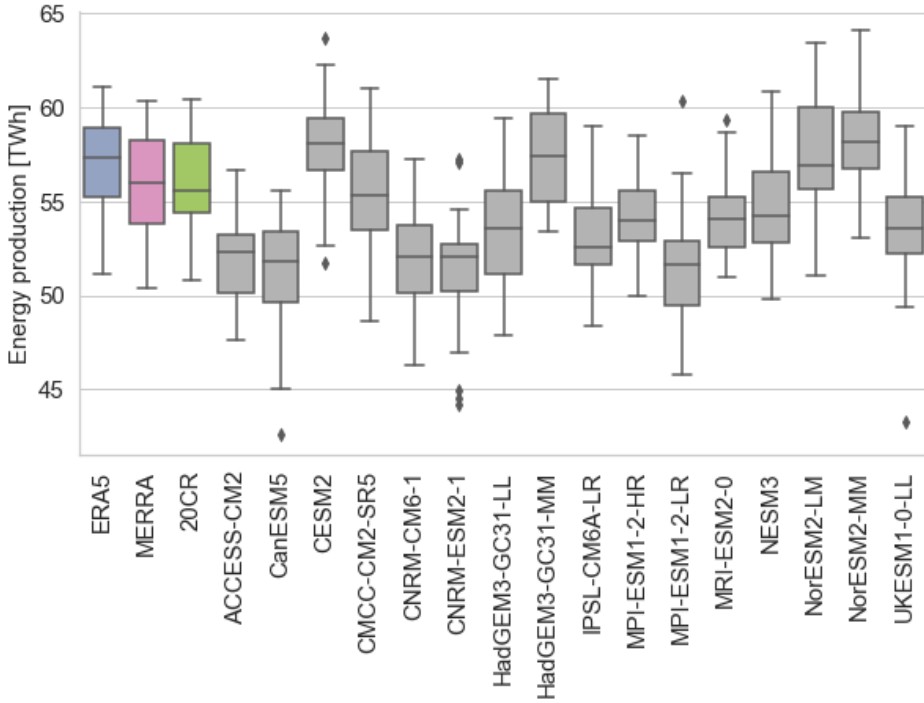

**Figure 7.** Net annual energy production of the Energy Island wind farm cluster calculated from the time series of the three reanalyses and the CMIP6 models (excluding the MIROC models) for the full historical period (1980–2014).

The net AEP derived from the CMIP6 models is, on average $4.9\,\%$ lower than that computed from the ERA5 data. Only three of the CMIP6 models have positive biases, and the ratio of the net AEP from CMIP6 and that calculated from the ERA5 time series varies between $-10\,\%$ and $2.4\,\%$. The AEP computed from the HadGEM3-GC31-MM model is $0.27\,\%$ of that calculated

using the ERA5 time series, not surprisingly since the wind climatology of this model is so close to that of the ERA5 (Fig. 6. The results of the net AEP comparison are consistent with the validation of the wind speed climatology in Fig. 2. The IQR of the net AEP from the CMIP6 is comparable to that from the reanalyses. Considering the nature of the CMIP6 models, these discrepancies are considered well within the expected uncertainty.

## 5    The future of wind resources in the North Sea

Based on the verification statistics presented, we focus on the changes in the future changes in the wind climatology using 16 of the 18 models in Table 1. The two CMIP6 simulations using the MIROC models are considered too low compared to the observed historical climatology in the North Sea and excluded from the analysis of the future wind resources.

We assess the changes in wind resources by first examining the large-scale changes in wind speed and wind power density over the whole northern Europe region. We consider two periods of 20 years each; the "past" from 1995 to 2014 and the



"future" from 2031 to 2050. Then we examine the wind speed trends in the North Sea by considering the entire 70 year period from 1980 to 2050. We finalise by evaluating the changes in wind speed and power density in the North Sea box and energy production at the Energy Island wind farm cluster from the past to the future.

## 5.1 Large scale changes

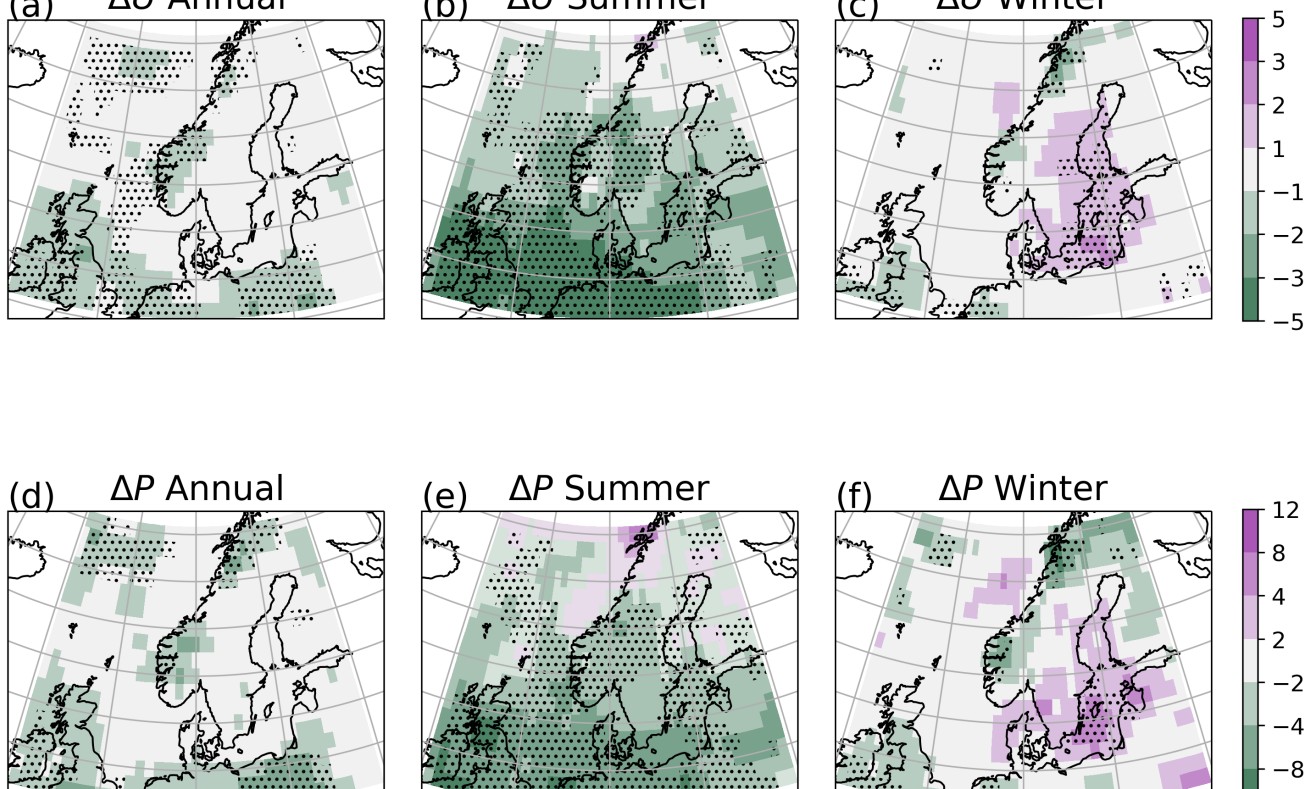

**Figure 8.** Median of the ensemble relative change [%] in 100 m AGL wind speed among the CMIP6 models [(a)–(c)] and wind power density [(d)–(f)] between the future (2031–2050) and the past (1995–2014). Medians are computed for the Annual means ((a) and (d)), JJA means [Summer, (b) and (e)], and DJF means [Winter, (c) and (f)]. The hatched areas represent areas where 75% (or 12 of 16) of models agree on the sign of the change.

The spatial distribution of the median relative changes in wind speed and wind power density is shown in Fig. 8. In the maps,

dotted areas represent those where over 75 % of the models (or 12 of the 16 models) agree on the sign of the change. This way of displaying agreement among the ensemble of models is a standard practice in IPCC assessment reports (IPCC, 2018, 2021). The maps are also made for summer and winter separately. The median of the annual changes at 100 m AGL are primarily





negative, minor (1–2 % in wind speed, 2–4 % in wind power density) and spread in patchy areas where the models disagree on the sign of the changes. During summer, however, median $100\,\mathrm{m}$ AGL wind speeds are reduced in the future by 3–5 % in

a spatially broad area covering the Southern North Sea, the British Islands and Northern Germany. Similar patterns but larger relative values are seen in the wind power density, with 5–15% reductions. In winter, the Baltic Sea experiences small but regionally consistent increases in wind speed (1–3 %) and power density (2–8 %) in the future, which are of the same sign in 12 or more models.

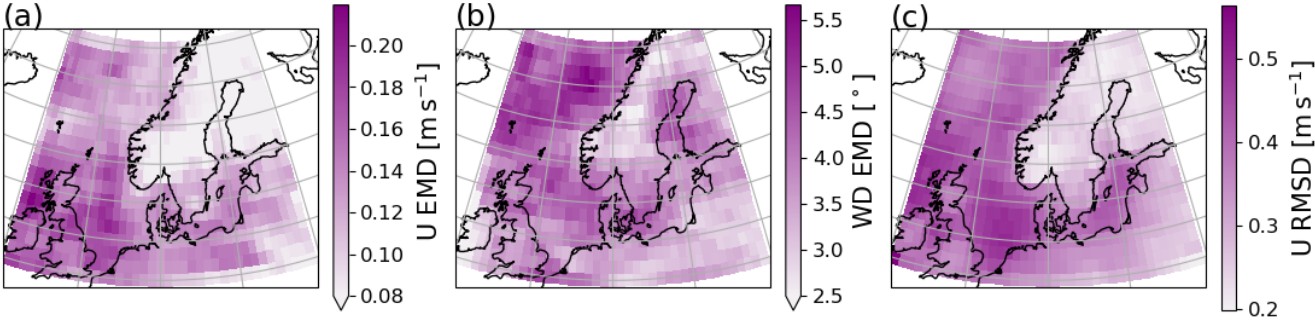

**Figure 9.** Median of the ensemble of (a) EMD of the $100\,\mathrm{m}$ AGL wind speed $[\mathrm{m\,s^{-1}}]$ and (b) CEMD of the $100\,\mathrm{m}$ AGL wind direction [°], and (c) RMSD of the annual cycle $[\mathrm{m\,s^{-1}}]$ between the past (1995–2014) and future (2031–2050) in the various CMIP6 models.

Figure 9 shows the maps of the median EMD, CEMD, and RMSD of the annual cycle between the past and future periods.

It is interesting to see the most significant changes in EMD concentrating in the middle of the North Sea and East of Scotland, with the lowest values over the Scandinavian peninsula. The median of the CEMD map (Fig. 9b) shows changes in the wind direction distribution with larger values in the north Atlantic off the Norwegian coast. These changes in wind direction could be relevant to wind farm calculations, as will be demonstrated in the next section for the future Energy Island wind farm. The monthly mean RMSD shows the changes in the annual cycle of the monthly mean wind. The figure shows larger values over

water than land, especially in the North Sea. This region and that over the South Baltic Sea agree with the assessment that seasonal changes in wind resources might be significant in this area.

## 5.2   Wind speed trends

Figure 10 shows the time series of annual mean wind speed averaged over the North Sea box delimited in the map in Fig. 1a. The thin background lines show each CMIP6 model (except for the MIROC models), and the coloured lines show the values

for the reanalyses in the historical period. The annual mean wind speed in the reanalyses shows significant variations from year to year, with a standard deviation of $0.33\,\mathrm{m\,s^{-1}}$, with the lowest value of the annual mean wind speed in 2010 of $9.3\,\mathrm{m\,s^{-1}}$ and the largest in 1990 of $10.8\,\mathrm{m\,s^{-1}}$ according to the ERA5 reanalysis.

The mean wind speed time series for the CMIP6 models also show significant interannual variations. The median of the CMIP6 models for the historical period lies within the envelope observed by the reanalyses, with a slight overestimation.



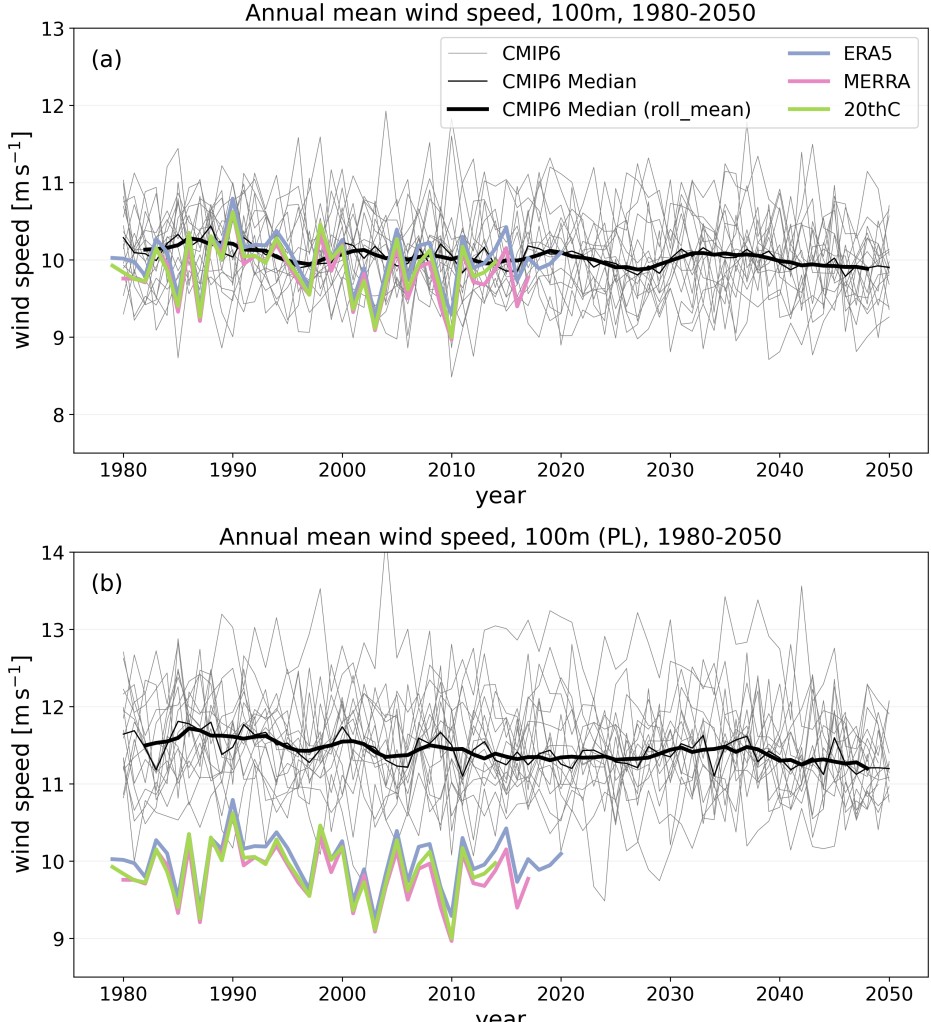

**Figure 10.** Annual mean wind speed at 100 m AGL using (a) model level winds and (b) a power-law relationship (Eq. 4) averaged over the box in Fig. 1a in the CMIP6 models. Also shown are annual mean of the three reanalyses (colours) and the median of all CMIP6 models and 5-year rolling mean of the median.





When looking at the full 70-year record (1980–2050), the median of the CMIP6 models shows slow decadal variations with a trend of $-0.032\,\mathrm{m\,s^{-1}}$ per decade, equivalent to $-0.32\,\%$ per decade. When examining the models individually, 3 of the 16 CMIP6 models show increases of 0.1% per decade or more, while 12 show decreases of $-0.1\,\%$ per decade or more. However, only 5 of the 15 models show significant downward trends with a $99\,\%$ significance level based on a Wald test with the Student t-distribution.

Several previous studies have used the $10\,\mathrm{m}$ AGL wind output to estimate the $100\,\mathrm{m}$ AGL wind speed using the power law (Eq. 4), sometimes with a different shear exponent $\alpha$ over the oceans. If used with $\alpha = 1/7$, the time series of annual mean wind speed look like those in Fig. 10b; such relation highly overestimates the turbine-height wind speed over water. While the mean in the reanalyses is $10\,\mathrm{m\,s^{-1}}$, that of the power-law-derived winds during the historical period is $11.4\,\mathrm{m\,s^{-1}}$, or 12% larger. In addition, the power-law extrapolated wind speeds exaggerate the trends in wind speed. In the extrapolated winds, the
trend in wind speed during 1980–2050 is $-0.41\,\mathrm{m\,s^{-1}}$ per decade, $28\,\%$ larger than that obtained by using the model-computed $100\,\mathrm{m}$ winds. Additionally, the significance level of the trends is enhanced, with now 8 of the 16 models having significant downward trends (with a $99\,\%$ significance level based on a Wald test) in wind speed.

### 5.3 Inter-model spread and annual cycle

The changes between the past (1995–2014) and the future (2031–2050) in mean wind speed and wind power density in the
CMIP6 models for the North Sea box (Fig. 1) are detailed in Fig. 11. We show the changes for the annual mean and four seasons: DJF (winter; December–February), MAM (spring; March–May), JJA (summer; June–August) and SON (autumn, September–November). Changes in annual mean wind speed have a small negative median and large spread. Only three of the 16 CMIP6 models have changes that are significant at the $95\,\%$ level (Fig. A1). Similar conclusions, but with an even larger spread, can be drawn for the changes in annual mean power density that vary from $-10$ to $8\,\%$. Using the CMIP model-derived
air density instead of standard atmosphere values does not significantly alter the annual mean wind power density results. Still, it narrows down the range of the relative differences.

The results are different when values are averaged over separate seasons. The decreases in the JJA mean wind speed are significant in half the models with a range of 4 to $6\,\%$ (or 0.4 to $0.6\,\mathrm{m\,s^{-1}}$). Many models also have significant changes in power density during the summer, with maximum decreases in the order of $15\,\%$. Only one model has a positive increase in
summer wind speed and power density over the North Sea. The changes in surface air density are nearly significant for all models but of no consistent sign among the models. Since changes in temperature due to climate change are significant in this region (IPCC, 2021), surface pressure must also play a significant role. However, their impact on the wind power density is mostly not significant.

Figure 11b shows the EMD calculated between the past (1995–2014) and future (2031–2050) wind speed distributions in
the 16 selected CMIP6 models. Changes in EMD vary between 0.07 to $0.34\,\mathrm{m\,s^{-1}}$, which is comparable to the values in the historical period when compared to the atmospheric reanalyses (Fig. 5). Larger EMD values are seen for all seasons, especially in summer. The spread among ensemble members is also much larger for the individual seasons, particularly in winter; however, this is partly an artefact of the reduced sample size. The wind direction CEMD between the past and the future is shown in





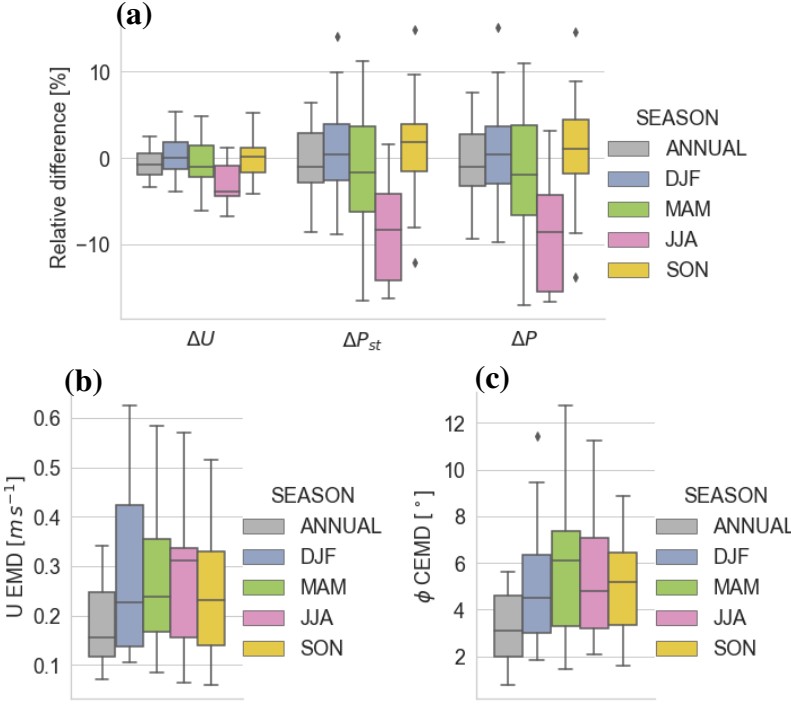

**Figure 11.** (a) Boxplot of the relative mean changes between the past (1995–2014) and the future (2031–2050) averaged over the North Sea box in Fig. 1a in $100\,\mathrm{m}$ AGL wind speed [$\Delta U$], standard wind power density [$\Delta P_{ST}$] and wind power density [$\Delta P$]. Boxplot of the changes in distribution between the past (1995–2014) and the future (2031–2050) as (b) wind speed EMD, and (c) wind direction CEMD. The boxes show the spread among the 16 CMIP6 models (excluding MIROC).

Fig. 11c. The wind direction CEMD is between $0.5$ to $5.6\,°$ in the annual wind distributions, with a median of $3°$. The wind
direction CEMD is larger for the individual seasons, especially during the spring (median of $6°$), when the spread is also the largest.

## 6   Energy production for the Energy Island wind farm cluster

We calculate the energy production of the wind farm cluster on the proposed Energy Island. The changes in EP (reference, standard and net) between the past (1995–2014) and the future (2031–2050) are presented as boxplots in Fig. 12 for the whole
year and the four individual seasons. The median reference AEP of the wind farm cluster is expected to decrease by about $-1.3\,\%$, as anticipated by the changes in wind speed. When we consider wakes, the median decrease in AEP is $-1.5\,\%$, and when we consider wakes and changes in air density, the median decline is $-1.9\,\%$. The spread among the CMIP6 models is large (with an IQR of 2.9 to 3.6 %) and increases from the reference value when wakes and density changes are considered.





The capacity factor of the wind farm cluster averaged over all CMIP6 models is reduced from $57.6\%$ in the past to $56.8\%$ in

the future.

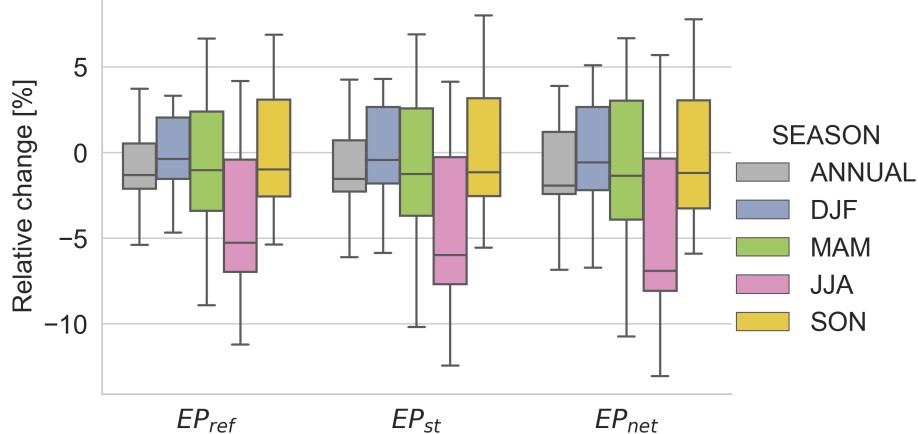

**Figure 12.** Boxplot of the change in energy production of the future Energy Island wind farm complex considering: reference EP (no wake loses), standard EP (standard air density), and net EP (including wake loses and air density). The changes are computed between the past (1995–2014) and the future (2031–2050) in the various CMIP6 models (excluding MIROC).

According to the changes in seasonal net EP between the past and the future, the median of all CMIP6 models is expected to decrease in all seasons, but very little in winter ($-0.6\%$) and markedly more in the summer ($-6.9\%$). The IQR of the changes in net EP also increases from winter ($4.9\%$) to summer ($7.7\%$). Similar trends in the capacity factors are calculated using the net seasonal EP (not shown).

In all the energy production estimates, the median change in $EP_{r}ef$ is decreased by including the wake losses. Including the effect of the changes in air density further reduces the energy production in the future in most models. For example, the summer energy production is predicted to change by $-5.3\%$, in the reference calculation, and $-6.0\%$, and $-6.9\%$, when wake losses and air density changes are considered, respectively. The effect of the decreases in the overall energy production (in the denominator) is not a primary factor. In the annual mean, median energy production is expected to decrease by $-18\,\mathrm{GW}$

when no wakes are considered but $-25\,\mathrm{GW}$ and $-33\,\mathrm{GW}$ when wakes and air density changes in the future are considered, respectively.

## 7   Summary and discussion

We show that most CMIP6 models can represent well the hub-height wind climatology of the historical period in the North Sea, especially considering the large model grid spacing. When using the 1980–2014 time series of wind speed and wind direction

and a hypothetical wind farm cluster, AEP calculations are within $10\%$ of that estimated using the wind time series extracted



from reanalyses data for the same period. The wind climate of the CMIP6 models is similar to that observed in this region in terms of wind direction and the phase and amplitude of the diurnal cycle. The CMIP6 simulation using the HadGEM3-GC31-MM model is as good quality as any of the reanalyses in representing the wind climate of the North Sea in the historical period.

Over the North Sea, we demonstrate that using the 10-m wind speed output from CMIP6 models with the power-law with $\alpha = 1/7$ overestimates the 100-m mean wind speed over the sea by 20 to 30%. This extrapolation will accordingly also exaggerate trends in wind speed and their significance. The wind shear in the boundary layer over the sea is generally smaller than that over the land (Peña and Hahmann, 2012; Hahmann et al., 2015) because of the low surface roughness length. But because the wind shear also depends on atmospheric stability, which could change, e.g., as the surface warms, the power-law with a

constant shear exponent becomes even less accurate for determining turbine-height winds (Badger et al., 2016). Over land, the wind shear exponent in the boundary layer is very variable (with typical values of $\alpha = 0.16$ over grass to $\alpha = 0.28$ over forest) and depends on the heights chosen, surface roughness length and atmospheric stability (Emeis, 2018). Over non-homogeneous terrain, the wind shear and power-law exponent at a site are also directional quantities, imprinting in the wind profile at various heights of the upstream flow at varying distances.

The changes in wind resources in Northern Europe shown here are somewhat inconsistent with some of those provided by the CMIP5 and Euro-CORDEX data in previous studies. Using ensembles of CMIP5 model output of various compositions and sizes, studies generally show an increase in wind speed in Central and Northern Europe and decreases in Southern Europe (Reyers et al., 2016; Carvalho et al., 2017; Gonzalez et al., 2019; Karnauskas et al., 2018; Devis et al., 2018). This is consistent with the projected northward shift and eastward extension of the North Atlantic jet and storm track into Europe under climate

change (Zappa et al., 2015). Studies using Euro-CORDEX data, however, have more varied results. Tobin et al. (2016) used an ensemble of nine Euro-CORDEX models and showed that changes in the AEP of the future European wind farms would remain within ±5% across the 21st century. Moemken et al. (2018) used a different set of nine Euro-CORDEX models, which projected small changes in mean annual and winter power output for large parts of Europe in future decades, but a considerable decrease for summer. The two Euro-CORDEX studies used different treatments of the wind speeds (one extrapolation from

10-m, the other interpolation and bias correction). They used models that do not account for changes in land use included in the global climate model forcing their regional climate model simulations.

Carvalho et al. (2021) used 10-m wind extrapolation using ERA5-derived shear parameter of a large ensemble of CMIP6 models (including the ones used in this study) and concluded that future energy resources are expected to decline by 10–20 % in practically all of Europe. The study also remarks that this contrasts with CMIP5 studies, which show an increase in Northern

Europe. The analysis of the $850\,\mathrm{hPa}$ zonal wind in Oudar et al. (2020) also finds differences in response between CMIP5 and CMIP6 simulations in the position of the jets in the North Atlantic and western Europe region. Additionally, Gonzalez et al. (2019) isolated the components associated with "large-scale" atmospheric circulation changes in the CMIP5 simulations and found that likely two processes are responsible. The first is related to changes in the large-scale atmospheric circulation, while the second is likely more local connected to changes in the near-surface boundary layer. This conclusion further emphasises



our hypothesis that the 10-m wind extrapolation with constant shear is likely to miss some of the processes controlling the future changes in wind resources.

We use the layout and turbine characteristics of a hypothetical massive wind farm cluster in the North Sea to evaluate future changes in wind speed and wind direction distributions for energy production in this region. Our results show that the summer decreases in wind speed are amplified when converted to energy production and further strengthened when we include the
inter-turbine wakes and changes in air density. This is because changes in wind speed impact power production in different ways depending on the wind speed. A change in wind speed near the cut-in (usually $3$–$5\ \mathrm{m\,s^{-1}}$) of a power curve will change the energy production while the same decrease in the rated power section of the power curve will not. Therefore, changes in wind speed at lower wind speeds are amplified when considering energy production. The difference in energy production is thus sensitive to the power curve of the chosen turbine type. Devis et al. (2018) showed that the projected wind power changes
might vary by up to half of their magnitude, depending on the type of turbine and region of interest. In addition, since the wake of one wind turbine decreases the wind speed downstream, including wake effects can reduce the wind speed near cut-in and further amplify the response of the wind farm to changes in wind speed. Because the wind direction distribution at a site dictates the combined magnitude and direction of the turbine wakes, it is expected that changes in wind direction in the future could also influence the efficiency of future wind farms, especially if the wind farm layout is optimised for present wind conditions.
In the calculation of the energy production of the Energy Island wind farm cluster, the engineering model does not consider the influence of the wind farm on the large-scale atmospheric flow (e.g., Nygaard, 2014; Fischereit et al., 2022). Large wind farms and clusters of wind farms can further decrease the wind farm efficiency by $10$ to $30\%$ in offshore regions with moderate wind speeds and relatively wide turbine spacing (Volker et al., 2017) depending on the size of the wind farm. This aspect further impacts the response of a given wind farm cluster to future changes in the wind climate and should be considered in
future studies.

## 8 Conclusions

This study examines the future mid-century changes in hub-height wind speed and hub-height power density over Northern Europe using an ensemble of CMIP6 model output according to the historical and SSP5-8.5 scenario simulations. We found non-significant differences between the past (1995–2014) and the future (2031–2050) in the annual mean of these quantities.
However, over $75\%$ of the models agree on the decrease of the resources during the summer in the North Sea. In the Baltic Sea, over $75\%$ of the models agree on an increase in wind resources during winter. However, for these future predictions to be meaningful, including as many models as possible is vital because models often disagree, and randomly selecting models might produce very different results.

The large-scale characteristics of the wind resources in the North Sea at levels accessible to present wind turbines are well
represented by current reanalysis compared to long-term tall mast measurements. The CMIP6 models run for the historical period can also capture the main features of the wind resources in this region. The errors in mean wind speed in a few CMIP6 models run at relatively high resolution lie within the spread of the three reanalyses.





Extrapolating wind speeds from 10 meters to turbine height using a constant power law is a poor approximation in many circumstances and will often exaggerate the changes in wind resources in the future. This is relevant because the variables that control the value of the power-law exponent (i.e., surface roughness length linked to vegetation changes and atmospheric stability) are likely to change in the future.

The changes in wind energy production shown here are in the range of $5\,\%$ to $10\,\%$. They would most likely be considered inconsequential, especially because energy systems, including large wind farms, are often over-designed to warranty the delivery of electricity under all conditions. However, such changes represent a sizeable economic impact in the form of higher financing prices and revenue loss by the wind farm operator. Regarding the design of the future low-emission power system, it is convenient that the significant decrease in wind resources occurs during summer when the wind electricity shortfall can be replaced by solar-generated electricity. Still, the consequences of reduced resources and possible changes in variability to the power system operation are more complex and not included in this study. As with the example provided here, power system studies must also use the complete chain of models and consider carefully the effect of simplifications used and the model numerical imprint on the model-generated time series.

*Code availability.* We provide a few examples of the Python code used to search, extract and interpolate and locally write the winds in the CMIP6 data. The code resides in https://github.com/ahahmann/future-wind. Once the article is accepted for publication, it will be updated and tagged in Zenodo.

The wake calculations were performed using PyWake, available at https://github.com/DTUWindEnergy/PyWake

*Acknowledgements.* The following people and organisations have kindly provided the tall mast data used for the verification. FINO 1, 2, and 3 were supplied by German Federal Maritime And Hydrographic Agency (BSH). Ijmuiden data from the Meteorological Mast Ijmuiden were provided by the Energy Research Center of the Netherlands (ECN) and processed by Peter Kalverla from Wageningen University. Birgitte Rugaard Furevik at the Geophysics Institute, University of Bergen, kidly supplied the Ekofisk data. Høvsøre data were provided by the Technical University of Denmark (DTU). The ERA5 data were downloaded from ECWMF and Copernicus Climate Change Service Climate Data Store. MERRA2 data are downloaded from the Distributed Active Archive Center (GSFC DAAC), DOI: 10.5067/VJAFPLI1CSIV. The 20th Century Reanalysis V3 data provided by the NOAA/OAR/ESRL PSL, Boulder, Colorado, USA, from their Web site at https://psl.noaa.gov/data/gridded/data.20thC_ReanV3.html. Support for the Twentieth Century Reanalysis Project version 3 dataset is provided by the U.S. Department of Energy, Office of Science Biological and Environmental Research (BER), the National Oceanic and Atmospheric Administration Climate Program Office, and by the NOAA Physical Sciences Laboratory. The CMIP6 multi-model ensemble data were downloaded through the distributed data archive developed and operated by the Earth System Grid Federation (ESGF; https://esgf.llnl.gov/).

AH and AP acknowledge the support of the Danish Ministry of Foreign Affairs, administered by the Danida Fellowship Centre under the project "Multi-scale and Model-scale Evaluation of Wind Atlases" (MEWA). OGS was funded by the European Union Horizon 2020 research and innovation program under grant agreement no. 861291 as part of the Train2Wind Marie Skłodowska-Curie ITN (https://www.train2wind.eu/). The authors thank Sarah Pryor and Jake Badger for many fruitful discussions.



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

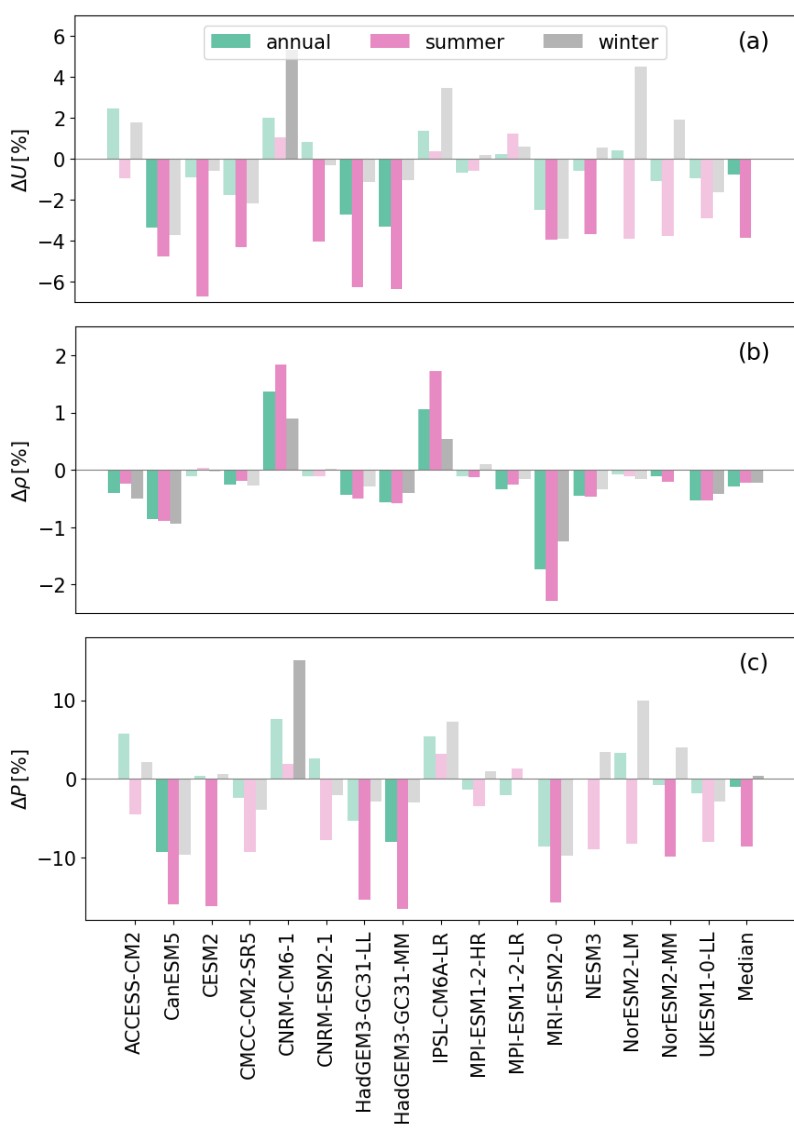

**Figure A1.** Relative changes in (a) wind speed, (b) surface air density, and (c) wind power density between the future and the past averaged over the North Sea box. Dark colour boxes are significant at the 95 %

.

*Author contributions.* AH wrote the first draft, extracted the CMIP6 data and analysed the results. OGS carried out the wind farm calculations. All authors participated in the design of the analysis and the writing and editing of the manuscript.

*Competing interests.* The authors declare that they have no conflict of interest.