# Peer review of "Current and future wind energy resources in the North Sea according to CMIP6"

_Wind Energy Science, 2022_

## Referee Comment (RC2)

The authors of this manuscript explored the current and future wind energy resources in Northern Europe using CMIP6 simulation. This work is interesting as it considered the wake effects on the annual energy production as the wake effects are ignored in all previous studies. The authors also assessed climate change for the 20 years 2031-2050, not at the end of the century, which is more important for the governments and wind farm developers. The subject discussed in the present article is of great importance, and the paper is well-structured and easy to follow. I truly congratulate the authors on this manuscript, which I believe should be considered for publication after minor changes.

The manuscripts, in my opinion, deserve revision on the following issues.

Q1. Paragraph 140. Did they use the wind speed of ERA5 reanalysis dataset? Or ERA5 forecast dataset? The hourly ERA5 reanalysis near-surface wind speed reveals a mismatch at 9:00-10:00 and 21:00-22:00 UTC (https://confluence.ecmwf.int/display/CKB/ERA5%3A+data+documentation), which shows an important impact on the research of diurnal cycle. ERA5 also suffers from a general underestimation bias of near-surface winds. Compared with the reanalysis dataset, the forecast near-surface winds show much better agreement between the assimilation cycles, at least on average.

Q2. The caption of Fig.8, also required more details that how to calculate the relative change between the future(2031-2050)and the past (1995-2014). The details would help the readers better understand.

Q3. Paragraph 395, "The wind climate of the CMIP6 models is similar to that observed in this region in terms of wind direction and the phase and amplitude of the diurnal cycle." However, the diurnal cycle was not compared and discussed in the manuscripts. It would be good if the authors could have some figures to compare the diurnal cycle of wind speed from the CMIP6 models and observations.

---

## Author Comment (AC1)

**Response to reviews – wes-2022-52**

Andrea N. Hahmann, Oscar García-Santiago and Alfredo Peña

October 4, 2022

**Reviewer #1**

Thank you for the positive evaluation of our manuscript. Your comments, together with those of referee #2, led to a slight revision of the paper. The specific changes are outlined below. In our response, the reviewers' comments are in black, and our responses are in blue.

The paper provides an estimate of the future changes in wind resource over the North Sea. To do so, a high-emission scenario is simulated in the CMIP6 models, which are first validated, in their historic portion, against reanalysis products, WRF simulations, and observations.

The topic is within the scope of WES. The paper is very well written, the methodology seems sound, and the standard of the figures is high enough for a peer-reviewed journal. In general, I do not have any major comments, so I would like to congratulate with the authors for a well-conducted analysis, and recommend publication of the paper after my few, very minor comments below have been addressed.

1. The last two sentences of the abstract seem to break the flow and are hard to connect to the rest of the abstract, I would suggest rephrasing.

   We have revised these two sentences. They now read:

   The common practice of extrapolating 10-meter wind speeds to turbine height using the power law with a constant shear exponent is often a poor approximation of the actual turbine height wind speed. This approximation can exaggerate the future changes in wind resources and ignores possible surface roughness and atmospheric stability changes.

2. 67: a capital letter "I" should be changed to "i".

   The text has been changed. Thank you.

3. 130: "to evaluate the boundary layer winds in the reanalysis datasets" is not clear since the reanalysis have not been introduced yet. Maybe it would be better to introduce the reanalysis first, and then the mast observations.

Thank you. Switching the order of the two data types makes sense and makes the manuscript flow better. We also changed the title of the subsection to "Reanalysis and mast observations"

4. 134: the sentence "We use the data from the 102 m." is not clear.

   Agreed. We have changed the sentence to read "We use data **from the anemometer** at $102\,\mathrm{m}$ **above mean sea level.**"

5. 139: can we really consider the NEWA a reanalysis product? As you state in Table 2, it does not assimilate any observations.

   Agreed. The NEWA dataset does not directly assimilate observations but uses spectral nudging to the driving reanalysis. We have changed the sentence that starts in L137 to read "We use the wind speed and direction derived from three modern reanalysis: ... ". We have also changed the headings in Table 2 from "Reanalysis product" to "Dataset"

6. Figure 4: missing "." at the end of the figure caption.

   Fixed.

7. 280: broken reference.

   Fixed.

8. 300: there is a repetition in this sentence, please correct.

   Fixed, thank you.

---

## Author Comment (AC2)

**Response to reviews – wes-2022-52**

Andrea N. Hahmann, Oscar García-Santiago and Alfredo Peña

October 4, 2022

**Reviewer #2**

Thank you for the positive evaluation of our manuscript. Your comments, together with those of referee #1, led to a slight revision of the paper. The specific changes are outlined below. In our response, the reviewers' comments are in black, and our responses are in blue.

The authors of this manuscript explored the current and future wind energy resources in Northern Europe using CMIP6 simulation. This work is interesting as it considered the wake effects on the annual energy production as the wake effects are ignored in all previous studies. The authors also assessed climate change for the 20 years 2031-2050, not at the end of the century, which is more important for the governments and wind farm developers. The subject discussed in the present article is of great importance, and the paper is well-structured and easy to follow. I truly congratulate the authors on this manuscript, which I believe should be considered for publication after minor changes.

The manuscript, in my opinion, deserves revision on the following issues.

Q1 Paragraph 140. Did they use the wind speed of ERA5 reanalysis dataset? Or ERA5 forecast dataset? The hourly ERA5 reanalysis near-surface wind speed reveals a mismatch at 9:00-10:00 and 21:00-22:00 UTC (`https://confluence.ecmwf.int/display/CKB/ERA5%3A+data+documentation`), which shows an important impact on the research of diurnal cycle. ERA5 also suffers from a general underestimation bias of near-surface winds. Compared with the reanalysis dataset, the forecast near-surface winds show much better agreement between the assimilation cycles, at least on average.

Only the ERA5 reanalysis was used to evaluate the results. We are aware of the discontinuity problems with the ERA5 forecasts (e.g., Kalverla et al. (2020) `https://doi.org/10.1002/qj.3748`), and we use the 6-hourly reanalysis data in the comparisons presented in the manuscript.

Q2 The caption of Fig.8, also required more details that how to calculate the relative change between the future (2031-2050) and the past (1995-2014). The details would help the readers better understand.

Agreed. We have added the following explanation in the text after L309. The figure caption now contains the sentence, "See text for the procedure used to compute the data displayed."

The figures are constructed in the following manner. The annual mean wind speed and wind power density of each CMIP6 model are bi-linearly interpolated to a $2.5° \times 2.5°$ regular grid. We then compute the difference between the means of the periods (2031–2050) and (1995–2014) for each model. For each grid point, we can find the model distribution of the changes, and the median of the change is then displayed. The spread is used to assess the agreement among the 16 CMIP6 models.

In addition to explaining how the data plotted was obtained, we have reversed what is plotted for agreement. The figure now shows, "The hatched areas represent areas **where less than** 75% (or 12 of 16) of models agree on the sign of the change."

Q3 Paragraph 395, "The wind climate of the CMIP6 models is similar to that observed in this region in terms of wind direction and the phase and amplitude of the diurnal cycle." However, the diurnal cycle was not compared and discussed in the manuscripts. It would be good if the authors could have some figures to compare the diurnal cycle of wind speed from the CMIP6 models and observations.

Agreed. The mention of the daily cycle has been removed. L397 now reads, "The wind climate of the CMIP6 models is similar to that observed in this region in terms **of wind speed and wind direction.**" The CMIP6 models only have four outputs each day, which, in our opinion, is insufficient to describe the daily cycle accurately.